# Turning inward in difficulties: R&D human resource slack, technological diversification, and independent innovation

Huijuan Li[1,2], Yinfei Zhao[2], Yang Li[1], Yong Wang◉[1]*

**1** School of Economics and Business Administration, Chongqing University, Chongqing, China, **2** College of Economics & Management, Hubei University of Arts and Science, Xiangyang, Hubei Province, China

* wangyongkt@126.com

**Data Availability Statement:** All relevant data are within the manuscript and its Supporting Information files.

## Abstract

Independent innovation emphasizes the self-reliance and control of all key links. Slack resources within an organization, especially for innovation, are the critical resources that are controllable for independent innovation. However, existing research still lacks evidence on the areas of slack innovation resources and independent innovation for deeper exploration. This research addresses this gap by providing an empirical analysis of the relationship between R&D human resource slack and firms' independent innovation. Based on the unbalanced panel data of China's listed manufacturing firms for eleven years, this research explores the effects of R&D human resource slack on firms' independent innovation, the mediating mechanism of technological diversification, and the boundary effects of top management team functional heterogeneity. The results reveal that R&D human resource slack positively affects firms' independent innovation; R&D human resource slack can promote firms' independent innovation through related technological diversification, while the mediating effect of unrelated technological diversification is not statistically significant; the top management team functional heterogeneity strengthens the positive impact of R&D human resource slack on firm independent innovation.

## 1. Introduction

The increasingly volatile world political and economic situation and the shrinking of complex value chains have forced China's manufacturing industry into a passive situation of 'squeezing from both ends': the protracted US-China trade friction, and the accelerating catch-up of the emerging manufacturing countries. Scientific and technological innovation has become the core area and main concern of great powers. To address this issue, the Science and Technology Progress Law was enacted at the end of 2021 in China, emphasizing 'stimulating independent innovation' and 'enhancing independent innovation capacity' from a legislative perspective. As it is seen, independent innovation is proposed in order not to be constrained by others in the field of key core technologies, to get rid of excessive dependence on external resources, as well as to cope with the challenges of crisis.

**Funding:** The author(s) received no specific funding for this work.

**Competing interests:** The authors have declared that no competing interests exist.

Firm innovation cannot be separated from the effective allocation and utilization of internal resources, and thus the impact of slack resources on independent innovation within the organization cannot be ignored. Firms often accumulate large amounts of slack resources due to sustained development, or because of their initiative to counteract turbulent environments [1], or to cultivate multi-skilled employees [2]. Failure to absorb these slack resources and allocate them effectively, especially in an environment full of uncertainty [3], not only makes it difficult to realize their potential value but also creates a resource load.

Therefore, on the one hand, the reality is that firms have large amounts of slack resources, while on the other hand, they face external resource constraints and high risks when innovating on their own. However, research on the impact of slack resources on firms' independent innovation is still limited in at least two aspects. First, there is a lack of research on the unique characteristics of independent innovation and the internal resource-driven perspective. Second, existing studies have not delved into slack resources at a more granular level, specific to a particular function, and in conjunction with specific economic environments. These gaps show that research should extend into the two areas of specific slack resources and independent innovation, and then explore them at a deeper level.

Innovation slack resources can be seen as a refinement of existing theories of organizational slack [4]. From the perspective of the innovation function of slack resources, it can be uncovered that slack resources will promote the focus of the firm on the core problem [5]. R&D human resource slack is a subcategory of slack resources [6]. Given that the R&D manpower and material inputs involved in independent innovation mostly come from within the firm, while the risks and benefits of innovation are all borne by the firm [7]. It is of great significance to explore and utilize internal R&D human resource slack to promote independent innovation capability under the constraint of external resources.

Knowledge and technology required for independent innovation by firms are carried by R&D personnel, so it is reasonable to speculate that there may be a research path in which slack R&D human resources utilize technological diversification, which ultimately affects the independent innovation of firms. In the meantime, executives and top management teams in the organization are core factors in business development. Individual executive members in a top management team are heterogeneous in terms of many background characteristics such as occupation, and are important factors associated with firm growth and performance [8]. Then, it is necessary to see how top management team functional heterogeneity affect the effect of R&D human resource slack.

Based on the data of listed firms in China's manufacturing industry from 2010 to 2021, this study conducts high-dimensional fixed effect model analysis to explore the role of R&D human resource slack on firm independent innovation, incorporating technological diversification into the theoretical framework to analyze the possible mediating mechanism. The moderating effect of the top management team functional heterogeneity is also examined.

This research contributes to the literature in the following strands. Firstly, this study refines the 'subjectivity' and 'resistance' characteristics of independent innovation. Then, it builds a research framework for the linkage between the R&D human resource slack and firms' independent innovation, which is based on the extension of the two fields of slack resources and innovation. On the one hand, it deepens and broadens the literature of independent innovation. On the other hand, it is the refinement and depth of the research on the field of slack resources. From the perspective of innovation function, this study explores the unique impact of R&D human resource slack within the organization on independent innovation, revealing their potential value for the independent innovation of firms. The slack of R&D human resources at least quantitatively guarantees the independence of the supply side of R&D personnel, provides the reserve power for the construction of the R&D talent ladder, qualitatively

improves the overall R&D resilience of firms, and promotes the technological diversification of R&D personnel.

Secondly, technological diversification is included in the model of this study to analyze the mechanism. The specialized knowledge and skills of R&D human resources slack will promote knowledge sharing and value creation through technological diversification, and then promote the independent innovation of firms. By constructing the theoretical model of 'R&D human resource slack-technological diversification-firm independent innovation', we hope to open the theoretical 'black box' of innovation slack resources affecting firm independent innovation. Besides, by incorporating the top management team into the model and integrating the resource-based theory and the upper-echelon theory, we analyze the role of top executives in the relationship between slack resources for innovation and independent innovation, thus expanding the theoretical boundaries of the existing research.

This study also has some practical implications. First, probing deeper into the internal slack resources of firms and their potential value can not only make up for the difficulty of obtaining external resources, but also alleviate the energy-consuming pressure of independent innovation, and at the same time, avoid firms from treating the slack resources, which may otherwise be important for independent innovation, in a simple and brutal way. Second, this study also draws attention to the limitations of slack R&D human resources by discriminating its knowledge base into related technological and unrelated technological diversification. Therefore, firms are encouraged to improve the resource balance and reasonably allocate the slack R&D human resources or optimize the quality of slack R&D human resources, such as cultivating versatile professionals.

The remainder of this study is organized as follows: Theoretical basis and research hypotheses section develops hypotheses based on a review of the related theoretical background. Research and data methodology section describes the research methods. The Empirical results and analyses section reports the empirical results. Last part discusses theoretical and practical implications and revelation for future research.

## 2. Theoretical basis and research hypotheses

### 2.1 Effects of R&D human resource slack on independent innovation

Independent innovation is one of the critical innovation strategies, but it has distinctive characteristics. It is informed that independent innovation is an innovation activity that firms adopt to overcome technology difficulties, attain valuable R&D achievements, and gain business profits from the technological achievements by themselves [7]. Puranam & Srikanth [9] also proposed a similar idea for independent innovation. They show that experienced acquirers are better able to mitigate the damaging consequences of the loss of autonomy associated with mergers and integration. Although the authors do not explicitly refer to the concept of 'independent innovation', they do point to the content of the 'subjectivity' that independent innovation emphasizes. It was not until 2009 that the concept of 'Independent Innovation' was explicitly proposed by scholars [10]. This study was published in the year following the 2008 financial crisis in the United States, reflecting the 'resistance' characteristic of independent innovation, which is of particular significance and value in the face of a stressful economic environment.

However, when it comes to independent innovation, it seems that existing research has been conducted more from the perspective of external support, focusing on external influences and paying little attention to the particular impact that important internal resources may have on firms' independent innovation.

Altaf & Shah [11] and Mousa & Reed [4] argue that innovation slack resources can be viewed as the stock of resources within a firm that is used for innovation functions, such as underutilized personnel dedicated to development (e.g., R&D talent pool). Innovation slack resources can be utilized and allocated so that they can be used for firm innovation activities such as developing new products [5], investing in new processes, and discovering new markets. Existing research pays little attention to the role of R&D human resources slack on firm independent innovation.

Therefore, this study synthesizes scholars' discussions and mainly refers to Wang et al.'s [6] definition, which suggests that R&D human resource slack is an excess of R&D employees held by an organization relative to industry norms. Arguments This practice may increase costs, but it ensures effectiveness when demand increases and provides the expertise needed to experiment and develop innovative projects [12]. There is still no consensus on the impact of slack resources on firm innovation. Therefore, it is important not to stereotype R&D human resource slack as either good or bad, but rather to combine the theory with the practical scenarios of firms and study them specifically to draw more stable conclusions.

R&D human resource slack may affect firm independent innovation in the following ways:

Firstly, independent innovation emphasizes the independent characteristics of innovation. To guarantee the smooth progress of innovation and not be restricted by others, R&D human resources is the first and foremost. As resource-based theory suggests, using resources instead of owning resources leads to competitive advantage [13, 14]. Therefore, correctly identifying and utilizing available R&D human resource slack would make it accessible to professional knowledge and skills. With a shared vision, more contact, and a high degree of trust, intellectuals' willingness to share knowledge, including tacit knowledge, would increase within the organization [15]. These knowledge resources are usually challenging to obtain from external accesses, and in turn, new knowledge and new technologies are created through knowledge interaction and sharing [16]. In addition, independent innovation also underscores the control of innovation results. Compared with cooperative innovation, independent innovation involves more core-technology breakthroughs, which are often difficult to quickly obtain by shortcuts from relying on external sources, thus requiring local talent teams to focus on core-technology breakthroughs systematically. Available R&D human resource slack within the organization would become essential support for establishing a core-technology talent team, and their specialized knowledge reserve and technical potential are vital for core-technology research.

Secondly, independent innovation typically arises to resist external pressure. Slack R&D human resources can provide a rich reserve pool of talents, which not only improves the overall R&D resilience of firms but also stimulates healthy competition among R&D personnel, thus helping firms to resist the pressure of uncertainty in the external environment. On the contrary, when facing the pressure of the external environment, simple and brutal layoffs will create a turbulent internal work environment, undermining firms' ability to combat the stress of the external environment. Further, deploying R&D talents in the early stages of firms' exploration of the frontier or response to increasing competition creates a slack of R&D human resources, which may become an important cornerstone for forming an independent, innovative advantage in the later stages. The specialized knowledge and skills that R&D human resource slack possesses can help firms quickly adapt to changes in the external environment. For example, to the production end, when international trade disputes appear or when the strategic consideration of energy security requires firms to eliminate backward production capacity and promote the development of products in the new field, firms don't need to slow down the pace due to the lack of R&D personnel or even choose to lay off employees as a result of deterioration in the space for survival. To the market end, when there are sudden changes

in consumer demand, firms reserving a certain number of R & D staff in the early stage would carry out technological upgrading or product transformation easily. In addition, R&D human resource slack also avoids the unexpected discontinuation of innovation due to the departure of technicians [17]. All in all, investing slack R&D human resources in specific technological fields would guarantee the continuous progress of firm independent innovation.

Based on the above analysis, we develop the following hypothesis:

**H1:** *A significant positive relationship exists between R&D human resource slack and firm independent innovation.*

## 2.2 The mediating role of technological diversification

Ansoff [18] first introduced the concept of diversification into the business and management disciplines. He pointed out that diversification is used to compensate for technological obsolescence, to diversify risk, to utilize excess production capacity, to reinvest earnings, and so on. Diversification of resources gives rise to diversified strategies. In this study, we focus on the technological dimension embedded in human resources and pay attention to the mechanism of the role of technological diversification of firms in the relationship between the slack of R&D human resources and firms' independent innovation.

Technological Diversification (TD) is an important way of combining resources. There are various definitions of TD, and here we draw on the definition given by Miller, which states that TD refers to the organization's pursuit of a diverse portfolio of R&D and its depth of knowledge networks [19]. TD is generally subdivided into related technological diversification (RTD) and unrelated technological diversification (UTD) [20]. RTD can be defined as the extent to which a firm is diversified in the field of technology, while UTD is the diversification of the organization across a wide range of scientific and technological fields [21].

Related technology is a strong linkage between the basic knowledge of the organization. RTD gradually accumulates the organization's research and development capabilities through these linkages, which helps to enhance the overall level of related knowledge of the organization to derive benefits from the knowledge and to reduce the organizational costs; in turn, RTD also utilizes the scale effect with the high speed of technological discovery to reduce the overall cost of the organization [19].

Firstly, the presence of heterogeneous knowledge enriches the possibility of new combinations and avoids the information cocoon effect, thus increasing the probability of new ideas emerging. The firm's slack of R&D human resources will allow it to cross-fertilize ideas through knowledge spillovers between units [22], broadening the knowledge breadth. Intangible assets that flow between firms through technological diversification will realize a higher value than in technologically homogeneous firms [23].

Secondly, essential resources required for independent innovation often have high transfer costs across firm boundaries [24]. In contrast, technological diversification [25] extends the connectivity of knowledge networks [19]. It expands the scope of existing technologies [26], thereby reducing the cost of innovation.

R&D human resource slack consists of individuals with similar expertise bases and skills, who can use organizational learning to deepen the process of accumulating knowledge and skills and expand these expertise and skills into related technology areas [20] Through the implementation of RTD, organizations may use this to break through the differentiated scope of science and technology, which in turn mitigates the risk of self-research [27].

Thus, we propose the following hypotheses:

**H2:** *R&D human resource slack positively affects related technological diversification.*

**H3:** *Related technological diversification has a positive effect on firm independent innovation.*

**H4:** *Related technological diversification mediates the positive relationship between R&D human resource slack and firm independent innovation. Specifically, R&D human resource slack enhances firm independent innovation by promoting related technological diversification.*

Independent innovation often requires the participation of human resources across disciplines and technological fields to realize new knowledge breakthroughs. As the slack of R&D human resources accumulates, the probability of including specialized technical talent from interdisciplinary fields increases accordingly, thus increasing the degree of unrelated technological diversification. However, inputs in unrelated technology fields may lead to an increase in innovation complexity, resulting in a discount in the efficiency of utilizing the already limited resources, and may bring about higher complexity of coordination, communication, resource allocation, and other managerial challenges compared to related technological diversification [28, 29].

Based on the above inferences, we propose the following hypotheses:

**H5:** *R&D human resource slack positively affects unrelated technological diversification.*

**H6:** *Unrelated technological diversification negatively impacts firm independent innovation.*

**H7:** *Unrelated technological diversification mediates the positive correlation between R&D human resource slack and firm innovation. Specifically, R&D human resource slack promotes unrelated technological diversification, yet unrelated technological diversification inhibits firms' ability to innovate independently.*

## 2.3 The moderating role of TMT functional heterogeneity

TMT functional heterogeneity refers to the extent to which members of the top manager team (TMT) differ in terms of the occupational experiences they have undertaken before becoming executives of the organization [30]. This variable captures the knowledge boundaries of the TMT and the differences in expertise among TMT members. For example, executives with a background in research and technology functions will be more willing to drive new technologies and see technology as a source of competitive advantage [31]. Heterogeneous TMT members offer a broader range of knowledge, skills, and perspectives, and will be more sensitive and problem-solving minded when faced with complex and changing environmental situations.

The core view of the top echelon theory suggests that TMT characteristics and experience, among others, are key variables in determining firm outcomes; TMT characteristics affect their perspectives, which in turn affect creativity and firm innovation [32]. It has been shown that functional background heterogeneity through internal job rotation promotes effective decision-making by TMTs and facilitates firms' innovation output; furthermore, functional background heterogeneity equips TMTs with a broader cognitive scope and decision-making horizon, which facilitates firms' innovation output by analyzing complex problems from a variety of perspectives, thus contributing to firms' innovation performance [33]. However, it has also been shown that TMT functional background heterogeneity can cause reduced performance [34]. So, it is curious to see how TMT functional background heterogeneity plays a moderating role.

Firstly, as the allocator of slack resources [35], TMTs would draw on the multiple perspectives and keen insights into new opportunities accumulated from heterogeneous career backgrounds [36] that can quickly capitalize on R&D human resource slack and leverage the potential of their professional skills, thus contributing to the enhancement of independent innovation capabilities.

In addition, the functional background heterogeneity implies that TMT members have a certain breadth of knowledge, which helps them utilize a wide range of expertise when making decisions [37]. Providing scientific and reasonable decision-making solutions for independent innovation guarantees that the potential value of R&D human resources slack is fully utilized at the decision-making level.

Furthermore, the functional background heterogeneity indicates that TMTs have more extensive interpersonal networks and resources, which can not only help TMT members pay more attention to the latest development trends in the core technology field but also find professional R&D technological talents in the related fields more quickly and optimize the quality of R&D human resources slack by facilitating the allocation of R&D talent teams, thus enhancing the independent innovation capability of the firm.

Based on this, the following hypothesis is proposed:

**H8:** *TMT functional heterogeneity plays a moderating role between R&D human resource slack and firm independent innovation. Specifically, TMT functional heterogeneity exacerbates the contribution of R&D human resource slack to firm independent innovation.*

Then, we develop a research framework better to understand the interplay between variables (**Fig 1**).

## 3. Research and data methodology

### 3.1 Sample selection and data sources

we selected a data set based on Chinese-listed manufacturing firms from the period between 2010 and 2021. The patent data are from CNRDS, and other data are sorted through the databases of CSMAR and Wind, which are essential sources of information reflecting the financial statements of China's stock market and listed firms and are also authorized databases in

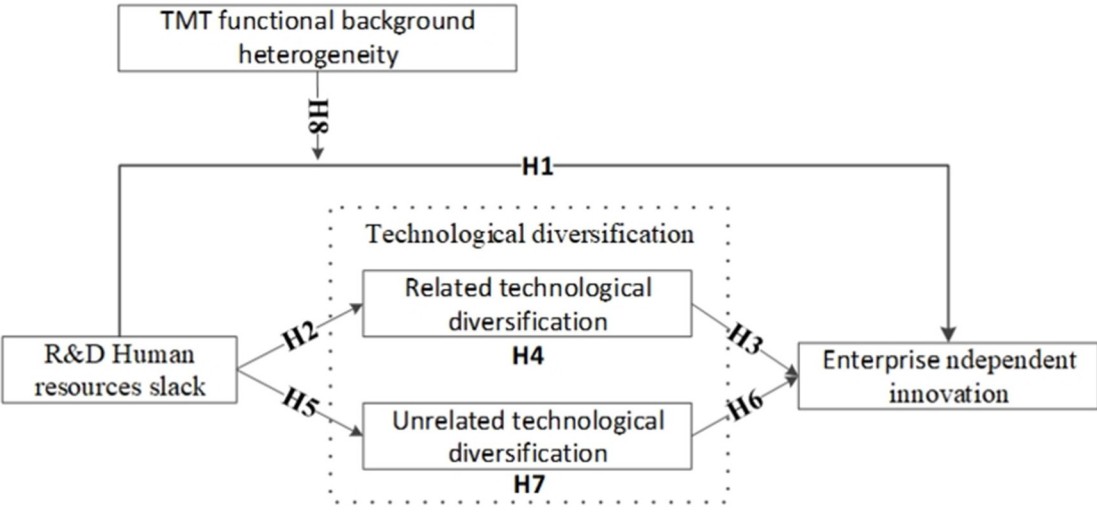

**Fig 1. Research framework.**

China. We constructed our research samples as the following steps: (1) FFCs and other diversified financial firms are excluded because of severe differences in their capital structure and regulatory environment compared to other firms; (2) Firms marked ST or stopped to be listed during the observation period are excluded; (3) Firms with missing financial data were excluded; (4) To avoid endogeneity problems, all variables except the dependent variable independent innovation are lagged by one period [38], i.e., variables such as R&D human resource slack, technological diversification, and TMT heterogeneity of the professional background have a period of 2012–2020, while the variable independent innovation has a period of 2013–2021; (5) To eliminate the effect of extreme values, all continuous variables were trimmed at the 1% level [39]. Ultimately, we obtained 4672 listed firms for eleven years of unbalanced panel data as empirical research samples.

## 3.2 Definition and measurement of variables

**3.2.1 Independent innovation (*II*).** According to the analysis above, independent innovation of firms has the subjectivity of 'independence and self-reliance' and the characteristic of 'pressure resistance'. It is an advanced stage of the technological innovation cycle of accumulation, from quantitative change to qualitative change. The main body of the independent innovation can dominate the patent and other innovation property rights. Therefore, drawing on Brockman et al. [40], the number of independently filed patents is used as a proxy variable to measure firm independent innovation.

First, patents filed independently by firms are more indicative of the 'self-reliant' subjectivity of independent innovation and the control of R&D results than patents filed jointly; Second, due to the heterogeneity and secrecy of independent innovation, it is difficult to trace and difficult to explore the process. Patents are probably the clearest measure of innovation [41] and are less subject to individual or subjective factors than other proxies typically measured using surveys (e.g., the number of new products or processes introduced by firms); Third, patents reflect the quality of the innovation. For an invention to be patented, it must be examined by experts who judge its novelty and utility. In contrast, reliable information on the quality of an innovation can rarely be gathered from other sources.

In addition, to deal with the right skewness of the patent data, all innovation indicators and patent applications filed by firms in a given year are Winsorized at 1% and 99% and then taken to the natural logarithm by adding 1.

**3.2.2 R&D human resource slack (*RHRS*).** Since the data in this paper covers all listed manufacturing firms, it is more appropriate to compare the focal firms to the industry-level R&D human resource information when calculating *RHRS*. Therefore, according to the findings of Lecuona & Reitzig and Paeleman & Vanacker [42, 43], this paper measures *RHRS* using information on employees in the R&D department, which is calculated as the ratio of the number of R&D personnel in a firm to the total number of employees in the firm, minus the median of the subindustry in which the firm is located for this indicator.

**3.2.3 Technological diversification (*TD*).** *TD* is derived from the idea of 'diversification of knowledge'. In the process of research and evolution, it is a reflection of the ability to provide technological support for the transformation and expansion of a firm through the process of expanding outward from the main areas of involvement, which is more often measured by the Herfindahl Index [44, 45]. *TD* is composed of *UTD* and *RTD*. Unrelated Technology Diversification (*UTD*) is the degree of diversification (i.e., a first-level patent class) in unrelated technology fields. Relevant Technological diversification (*RTD*) is the degree of diversification (i.e., secondary patent class) in relevant technology areas.

**3.2.4 Top management team functional heterogeneity (*FH*).** This research selects FH as a moderating variable to be measured. Referring to Michel & Hambrick [46], the Herfindal-Hirschman coefficient is used to categorize and measure *FH* [34].

Referring to Brochet et al. and Zhang et al. [47, 48], control variables are also constructed by considering potential factors that may affect *RHRS* and *II* of firms. The name and measurement of each variable are shown in **Table 1**.

## 3.3 Research models

This study adopts quantitative empirical analysis of relevant data of listed firms in the manufacturing industry, mainly through correlation analysis, regression analysis, and panel multiple regression analysis to test the relationship between R&D human resource slack and firm independent innovation. Referring to Zhou et al. [49], we construct the following high-dimensional fixed effect models to test our research hypotheses:

**Table 1. Definition and measurement of variables.**

| Variable name | Code | Index |
|---|---|---|
| Independent innovation | *II* | the number of independently filed patents is used as a proxy variable to measure firm independent innovation |
| R&D human resource slack | *RHRS* | Number of R&D employees in the firm / Total number of employees in the firm-Industry median of this indicator |
| Technological diversification | *TD* | $TD = \dfrac{1}{\sum\limits_{i} (N_i/N)^2}$, where $N_i$ represents the number of patents belonging to the I field in a certain firm's patents |
| Unrelated technological diversification | *UTD* | $UTD = \sum\limits_{j=1}^{M} P_j \ln\left(\dfrac{1}{P_j}\right)$, where $P_j$ is the ratio of patents in the first-level patent class j. There are M different first-level patent classes. |
| Related technological diversification | *RTD* | RTD = TD-UTD |
| Top management team functional heterogeneity | *FH* | $FH = 1 - \sum\limits_{i=1}^{n} P_i^2$ Where $P_i$ denotes the proportion of category I members of the team to the total number of team members. H value is between 0–1, the larger the value, the larger the value of TMT functional heterogeneity. |
| Firm size | *Size* | Natural logarithm of total assets at year-end |
| Firm age | *FirmAge* | Current year minus the year of incorporation plus 1 in natural logarithms |
| Financial leverage | *Lev* | Total liabilities divided by total assets |
| Corporate Growth | *Growth* | Current year's operating income/previous year's operating income - 1 |
| State-owned firms | *SOE* | Dummy variable that takes the value of 1 if the firm is a state-owned firm; 0 otherwise |
| Tobin's Q | *TobinQ* | (Market value of outstanding shares + number of non-outstanding shares × net assets per share + book value of liabilities)/Total assets |
| Board size | *Board* | The number of board members is taken as a natural logarithm |
| Shareholding concentration | *Top10* | Number of shares held by top ten shareholders/total shares |
| Board independence | *Indep* | Independent directors divided by the number of directors |
| Shareholding checks and balances | *Balance* | the ratio of the sum of the shareholdings of the second through fifth-largest shareholders to the shareholding of the first largest shareholder |
| the duality of the Chair of the board and CEO | *Dual* | Dummy variable that takes the value of 1 when the chairman and CEO are the same person; 0 otherwise |
| Manager holdings | *Mhold* | the percentage of total shares held by managers to total shares outstanding |

*Model 1*: $Inno_{i,t} = \alpha + \beta_{1,1}RHRslack_{i,t-1} + \lambda CVs_{i,t} + \sum Year + \sum Ind + \varepsilon_{i,t-1}$

*Model 2*: $RTD_{i,t} = \alpha + \beta_{2,1}RHRslack_{i,t} + \lambda CVs_{i,t} + \sum Year + \sum Ind + \varepsilon_{i,t}$

*Model 3*: $UTD_{i,t} = \alpha + \beta_{3,1}RHRslack_{i,t} + \lambda CVs_{i,t} + \sum Year + \sum Ind + \varepsilon_{i,t}$

*Model 4*:

$$Iinno_{i,t} = \alpha + \beta_{4,1}RTD_{i,t-1} + \beta_{4,2}RHRslack_{i,t-1} + \lambda CVs_{i,t-1} + \sum Year + \sum Ind + \varepsilon_{i,t-1}$$

*Model 5*:

$$Iinno_{i,t} = \alpha + \beta_{5,1}UTD_{i,t-1} + \beta_{5,2}RHRslack_{i,t-1} + \lambda CVs_{i,t-1} + \sum Year + \sum Ind + \varepsilon_{i,t-1}$$

*Model 6*:
$$Iinno_{i,t} = \alpha + \beta_{6,1}RHRslack_{i,t-1} \times Difjob_{i,t-1} + \beta_{6,2}RHRslack_{i,t-1}$$
$$+ \beta_{6,3}Difjob_{i,t-1} + \lambda CVs_{i,t-1} + \sum Year + \sum Ind + \varepsilon_{i,t-1}$$

where $\alpha$ is the intercept term, $\beta_\gamma$ ($\gamma = 1, \ldots, 6$) is the coefficient, $\lambda$ is the coefficient and $\varepsilon$ is the random error term.

*Model 1* is a regression model of the independent variable R&D human resource slack and the dependent variable firm independent innovation used to test *H1*.

*Model 2(3)* is a regression model of R&D human resource slack and related (unrelated) technological diversification, which is used to test the mediating role of technological diversification, including *H2(5)*.

*Model 4(5)* adds related (unrelated) technological diversification based on *model 1*, which is used to test *H3/4 (6/7)*, that is, the relevant technological diversification mediating the relationship between slack of R&D human resource and independent innovation of firms.

*Model 6* adds the TMT functional heterogeneity to test *H8*.

We use robust standard errors to control the effect of heteroskedasticity in the error terms and correlation issues in the time series on the standard errors of the estimated coefficients. In addition, the respective variables are centered to minimize multicollinearity [50].

## 4. The empirical results and analyses

### 4.1 Variable description

**Table 2** provides the results of the descriptive statistics analysis for each of the main variables in this study. The total sample size is 4672 observations, and the descriptive statistics of the variables are generally consistent with existing studies.

### 4.2 Correlation analysis

**Table 3** provides the results of the Pearson correlation analysis of variables. There is a strong correlation between the slack of R&D human resources and independent innovation. Next, the variance inflation factor (*VIF*) analysis is conducted. The results show that the mean value of VIF of all model variables is less than 3. The maximum value of VIF of each variable is 2.1, which is much less than 10, indicating that the problem of multicollinearity has been effectively controlled. The accuracy of the regression results can be effectively ensured.

### 4.3 Multiple regression results and discussion

**4.3.1 A test of the relationship between R&D human resource slack and technological diversification.** The results of the empirical analysis of the regression model of R&D human resources slack and technological diversification are shown in **Table 4**.

First, the relationship between R&D human resource slack (RHRS) and related technological diversification (RTD) is analyzed. Among them, *column (I)* is the regression of the control

**Table 2. Descriptive statistics of variables.**

| Variable name | Mean | SD | Min | Median | Max. |
|---|---|---|---|---|---|
| Inno | 2.976 | 1.166 | 1.099 | 2.833 | 5.864 |
| RHRS | 0.003 | 0.096 | -0.189 | -0.019 | 0.714 |
| RTD | -0.003 | 0.474 | -0.736 | -0.053 | 1.201 |
| UTD | -0.002 | 0.501 | -1.126 | 0.028 | 0.948 |
| FH | 0.001 | 0.079 | -0.709 | 0.016 | 0.134 |
| Size | 22.130 | 1.146 | 19.872 | 21.975 | 26.398 |
| FirmAge | 2.923 | 0.267 | 2.197 | 2.944 | 3.555 |
| Lev | 0.379 | 0.170 | 0.055 | 0.376 | 0.906 |
| Growth | 0.169 | 0.294 | -0.660 | 0.125 | 3.335 |
| SOE | 0.201 | 0.401 | 0.000 | 0.000 | 1.000 |
| TobinQ | 2.026 | 1.163 | 0.802 | 1.688 | 11.393 |
| Board | 2.106 | 0.188 | 1.609 | 2.197 | 2.708 |
| Top10 | 0.592 | 0.138 | 0.236 | 0.600 | 0.910 |
| Indep | 0.376 | 0.054 | 0.308 | 0.333 | 0.571 |
| Balance | 0.818 | 0.631 | 0.036 | 0.652 | 2.961 |
| Dual | 0.336 | 0.472 | 0.000 | 0.000 | 1.000 |
| Mhold | 0.192 | 0.208 | 0.000 | 0.110 | 0.697 |

Note: * indicates p < 0.100

** indicates p < 0.050

*** indicates p < 0.010

variables on the dependent variable RTD. *Column (II)* is the regression after adding the independent variable RHRS to *column (I)*, that is *Model 2*. It can be seen that the regression coefficient of RHRS and RTD is 1.037, and it is significant at a 1% level, which indicates that as the level of RHRS increases, the level of RTD increases. **H2** is supported.

Next, the relationship between R&D human resource slack (RHRS) and unrelated technological diversification (UTD) is analyzed. *Column (III)* is the regression of control variables on UTD. *Column (IV)* is the regression after adding the independent variable RHRS to *column (III)*, that is *Model 3*. It can be learned that the regression coefficient of RHRS and UTD is not significant, which indicates that there is no significant correlation between RHRS and UTD. **H5** is **not** supported.

To verify the relationship between the slack of R&D human resources, technological diversification, and firm independent innovation, regression analysis is carried out on the relevant variables. The results of the empirical analysis of the regression model are shown in **Table 5**.

Among them, *column (V)* is the regression of the control variables on the dependent variable firm independent innovation (Inno). *Column (VI)* (i.e., *Model 1*) is the regression after including the independent variable R&D human resource slack (RHRS) in *column (V)* to test the direct effect of R&D human resource slack on the firm independent innovation. *Column (VII)* is a regression after including the squared term of R&D human resource slack (RHRS$^2$) in *column (VI)* to verify whether there is a nonlinear relationship between R&D human resource slack and firm independent innovation. *Column (VIII)* (i.e., *Model 4*) is the regression result after including related technological diversification (RTD) into *column (VI)* to test the mediating role of related technological diversification between R&D human resource slack and firm independent innovation. *Column (IX)* (i.e., *Model 5*) is the regression result after incorporating unrelated technological diversification (UTD) into *column (VI)* to test the

**Table 3. Pearson correlation analysis of variables.**

| Variable name | Iinno | RHRS | RTD | UTD | FH | Size | FirmAge |
|---|---|---|---|---|---|---|---|
| Iinno | 1.000 | | | | | | |
| RHRS | 0.066*** | 1.000 | | | | | |
| RTD | 0.433*** | 0.182*** | 1.000 | | | | |
| UTD | 0.283*** | -0.057*** | 0.417*** | 1.000 | | | |
| FH | -0.007 | 0.013 | 0.079*** | 0.060*** | 1.000 | | |
| Size | 0.368*** | -0.117*** | 0.466*** | 0.459*** | 0.028* | 1.000 | |
| FirmAge | -0.009 | -0.116*** | 0.045*** | 0.117*** | 0.067*** | 0.183*** | 1.000 |
| Lev | 0.206*** | -0.133*** | 0.314*** | 0.313*** | 0.013 | 0.518*** | 0.076*** |
| Growth | 0.047*** | 0.025* | 0.004 | 0.010 | -0.018 | 0.021 | -0.145*** |
| SOE | 0.117*** | -0.054*** | 0.186*** | 0.177*** | 0.070*** | 0.339*** | 0.233*** |
| TobinQ | -0.052*** | 0.155*** | -0.125*** | -0.147*** | -0.008 | -0.200*** | -0.078*** |
| Board | 0.095*** | -0.089*** | 0.123*** | 0.171*** | 0.071*** | 0.257*** | 0.143*** |
| Top10 | 0.063*** | -0.063*** | -0.120*** | -0.082*** | -0.052*** | -0.049*** | -0.123*** |
| Indep | 0.052*** | 0.032** | 0.004 | -0.019 | -0.074*** | 0.022 | -0.029* |
| Balance | 0.009 | 0.105*** | -0.032** | -0.072*** | 0.026* | -0.089*** | -0.030** |
| Dual | -0.027* | 0.088*** | -0.065*** | -0.100*** | -0.047*** | -0.156*** | -0.117*** |
| Mhold | -0.077*** | 0.102*** | -0.157*** | -0.188*** | -0.026* | -0.389*** | -0.187*** |
| **Variable name** | **Lev** | **Growth** | **SOE** | **TobinQ** | **Board** | **Top10** | **Indep** |
| Lev | 1.000 | | | | | | |
| Growth | 0.078*** | 1.000 | | | | | |
| SOE | 0.203*** | -0.080*** | 1.000 | | | | |
| TobinQ | -0.281*** | 0.093*** | -0.070*** | 1.000 | | | |
| Board | 0.120*** | -0.012 | 0.254*** | -0.028* | 1.000 | | |
| Top10 | -0.114*** | 0.067*** | -0.134*** | 0.057*** | -0.111*** | 1.000 | |
| Indep | 0.013 | -0.001 | -0.039*** | -0.019 | -0.577*** | 0.085*** | 1.000 |
| Balance | -0.055*** | 0.030** | -0.196*** | -0.016 | 0.027* | 0.002 | -0.016 |
| Dual | -0.079*** | 0.045*** | -0.269*** | 0.045*** | -0.158*** | 0.085*** | 0.114*** |
| Mhold | -0.216*** | 0.104*** | -0.422*** | 0.007 | -0.220*** | 0.254*** | 0.066*** |
| **Variable name** | **Balance** | **Dual** | **Mhold** | | | | |
| Balance | 1.000 | | | | | | |
| Dual | 0.022 | 1.000 | | | | | |
| Mhold | 0.200*** | 0.204*** | 1.000 | | | | |

Note:

* indicates $p < 0.100$

** indicates $p < 0.050$

*** indicates $p < 0.010$

mediating role of unrelated technological diversification between R&D human resource slack and firm innovation.

From the regression results of *column (VI)* (i.e., *Model 1*), it can be seen that the regression coefficient of R&D human resources slack and firm independent innovation is 0.970. It is significant at a 1% level, which indicates that the independent innovation ability is increased with the increase of R&D human resources slack. There is a linear positive correlation between the two. From the regression results of *column (VII)*, it can be seen that the correlation coefficient between R&D human resource slack and firm independent innovation is 0.901. It is significant at a 1% level, but the regression coefficient of the squared term of R&D human resource slack

**Table 4. Results of regression analysis of R&D human resource slack affecting technological diversification.**

| Model variable name | (I) RTD | (II) M2 RTD | (III) UTD | (IV) M3 UTD |
|---|---|---|---|---|
| Size | 0.200*** | 0.200*** | 0.179*** | 0.179*** |
| | (0.006) | (0.006) | (0.007) | (0.007) |
| FirmAge | -0.052** | -0.019 | 0.001 | 0.004 |
| | (0.022) | (0.022) | (0.026) | (0.026) |
| Lev | 0.035 | 0.095** | 0.126*** | 0.131*** |
| | (0.040) | (0.039) | (0.045) | (0.045) |
| Growth | -0.010 | -0.012 | 0.015 | 0.015 |
| | (0.019) | (0.019) | (0.023) | (0.023) |
| SOE | 0.092*** | 0.076*** | 0.016 | 0.015 |
| | (0.017) | (0.016) | (0.018) | (0.019) |
| TobinQ | 0.004 | -0.008* | -0.006 | -0.007 |
| | (0.005) | (0.005) | (0.006) | (0.007) |
| Board | 0.104*** | 0.135*** | 0.210*** | 0.212*** |
| | (0.037) | (0.035) | (0.044) | (0.045) |
| Top10 | -0.252*** | -0.174*** | -0.194*** | -0.188*** |
| | (0.043) | (0.042) | (0.049) | (0.050) |
| Indep | 0.100 | 0.098 | 0.248* | 0.248* |
| | (0.121) | (0.117) | (0.142) | (0.142) |
| Balance | -0.011 | -0.026*** | -0.028*** | -0.029*** |
| | (0.009) | (0.008) | (0.010) | (0.010) |
| Dual | 0.020 | 0.007 | -0.018 | -0.019 |
| | (0.012) | (0.012) | (0.014) | (0.014) |
| Mhold | 0.120*** | 0.084*** | 0.033 | 0.030 |
| | (0.031) | (0.030) | (0.038) | (0.038) |
| RHRS | | 1.037*** | | 0.085 |
| | | (0.059) | | (0.064) |
| _cons | -4.443*** | -4.624*** | -4.407*** | -4.422*** |
| | (0.157) | (0.149) | (0.190) | (0.191) |
| adj. $R^2$ | 0.408 | 0.448 | 0.286 | 0.286 |
| YEAR FE | Yes | Yes | Yes | Yes |
| IND FE | Yes | Yes | Yes | Yes |
| F | 167.523 | 192.814 | 115.394 | 106.615 |
| N | 4672 | 4672 | 4672 | 4672 |

Note:

* indicates $p < 0.100$

** indicates $p < 0.050$

*** indicates $p < 0.010$.

on firm independent innovation is not significant, which further indicates that R&D human resource slack is positively correlated with firm independent innovation. There is no curvilinear relationship between the two. Therefore, **H1** is supported.

In **Table 5**, the adjusted $R^2$ increases from 0.303 in *column (VI)* to 0.346 in *column (VIII)*, which indicates that the explanatory power of R&D human resource slack on the changes of firms' independent innovation increases when the mediating variable related technological

**Table 5. Results of regression analysis of the mediating role of technological diversification.**

| Model variable name | (V) Inno | (VI) M1 Inno | (VII) Inno | (VIII) M4 Inno | (IX) M5 Inno |
|---|---|---|---|---|---|
| Size | 0.446*** | 0.446*** | 0.446*** | 0.309*** | 0.384*** |
|  | (0.018) | (0.018) | (0.018) | (0.019) | (0.019) |
| FirmAge | -0.055 | -0.025 | -0.024 | -0.012 | -0.026 |
|  | (0.059) | (0.059) | (0.059) | (0.057) | (0.058) |
| Lev | -0.161 | -0.105 | -0.104 | -0.170* | -0.151 |
|  | (0.102) | (0.102) | (0.102) | (0.098) | (0.101) |
| Growth | -0.014 | -0.016 | -0.015 | -0.008 | -0.021 |
|  | (0.050) | (0.049) | (0.049) | (0.048) | (0.049) |
| SOE | 0.135*** | 0.120*** | 0.119*** | 0.067 | 0.114*** |
|  | (0.044) | (0.044) | (0.044) | (0.043) | (0.044) |
| TobinQ | 0.087*** | 0.076*** | 0.075*** | 0.081*** | 0.078*** |
|  | (0.015) | (0.014) | (0.014) | (0.014) | (0.014) |
| Board | 0.454*** | 0.483*** | 0.486*** | 0.390*** | 0.409*** |
|  | (0.101) | (0.101) | (0.101) | (0.098) | (0.100) |
| Top10 | 0.495*** | 0.568*** | 0.567*** | 0.687*** | 0.633*** |
|  | (0.117) | (0.118) | (0.118) | (0.113) | (0.116) |
| Indep | 1.174*** | 1.172*** | 1.178*** | 1.104*** | 1.085*** |
|  | (0.337) | (0.336) | (0.336) | (0.327) | (0.338) |
| Balance | 0.048** | 0.034 | 0.034 | 0.052** | 0.045* |
|  | (0.024) | (0.024) | (0.024) | (0.023) | (0.024) |
| Dual | 0.051 | 0.038 | 0.038 | 0.034 | 0.045 |
|  | (0.032) | (0.032) | (0.032) | (0.031) | (0.031) |
| Mhold | 0.290*** | 0.256*** | 0.256*** | 0.198** | 0.246*** |
|  | (0.083) | (0.082) | (0.082) | (0.080) | (0.081) |
| RHRS |  | 0.970*** | 0.901*** | 0.258 | 0.941*** |
|  |  | (0.159) | (0.212) | (0.158) | (0.156) |
| RHRS$^2$ |  |  | 0.365 |  |  |
|  |  |  | (0.687) |  |  |
| RTD |  |  |  | 0.687*** |  |
|  |  |  |  | (0.040) |  |
| UTD |  |  |  |  | 0.349*** |
|  |  |  |  |  | (0.033) |
| _cons | -8.681*** | -8.851*** | -8.847*** | -5.675*** | -7.307*** |
|  | (0.459) | (0.454) | (0.455) | (0.483) | (0.488) |
| adj. $R^2$ | 0.298 | 0.303 | 0.303 | 0.346 | 0.319 |
| YEAR FE | Yes | Yes | Yes | Yes | Yes |
| IND FE | Yes | Yes | Yes | Yes | Yes |
| F | 83.648 | 81.465 | 76.115 | 107.138 | 85.273 |
| N | 4672 | 4672 | 4672 | 4672 | 4672 |

Note:

* indicates $p < 0.100$

** indicates $p < 0.050$

*** indicates $p < 0.010$.

diversification is added, and related technological diversification may play a mediating role between the two, pending further analysis of the regression equation results.

**4.3.2 A test of the mediating role of related technological diversification.** According to Zhao et al. [51], the mediating role of related technological diversification is tested based on the following procedures and principles:

In the first step, it has been verified that the direct effect of the independent variable R&D human resources slack on the dependent variable firm independent innovation is significant (the regression coefficient is 0.970 and significant at a 1% level) through the *column (VI)* (i.e., *Model 1*) in **Table 5**. On this basis, considering the significance of the mediating role related to technological diversification.

In the second step, *column (II)* (i.e. *Model 2*) from **Table 4** has been previously verified to show a positive correlation between R&D human resource slack and related technological diversification (the regression coefficient is 1.037 and significant at a 1% level).

In the third step, the results of *column (VIII)* (i.e., *Model 4*) in **Table 5** show that the regression coefficient of related technological diversification and firm independent innovation is 0.687, and it is significant at a 1% level, related technological diversification, and firm independent innovation present a positive correlation, and the enhancement of related technological diversification will promote the enhancement of independent innovation ability. **H3** is supported. At this time, the coefficient of the slack of R&D human resources and firm independent innovation is not significant. Therefore, the related technological diversification completely mediates the positive correlation between R&D human resources slack and firm independent innovation. Thus, **H4** is supported.

**4.3.3 A test of the mediating role of unrelated technological diversification.** The mediating role of irrelevant technological diversification is verified next. *Column (IX)* in **Table 5** is the regression result after incorporating irrelevant technological diversification into the direct effect of *column (VI)* to test the mediating role of irrelevant technological diversification between R&D human resource slack and firm independent innovation. At this time, the adjusted $R^2$ rises from 0.303 in *column (VI)* to 0.319 in *column (IX)*, with an increase in explanatory power, suggesting that irrelevant technological diversification may play a mediating role between the relationship between R&D financial slack and firm independent innovation, pending further analysis of the regression results. In this paper, we refer to Wen et al. [52] to test the mediating role of irrelevant technological diversification according to the following procedures and principles:

In the first step, it has been verified that the direct effect of the independent variable R&D human resources slack on the dependent variable firm independent innovation is significant through *column (VI)* in **Table 5**;

In the second step, it has been verified that there is no significant correlation between the independent variable R&D human resource slack and the mediator variable irrelevant technological diversification through *column (IV)* (i.e. *Model 3*) in **Table 4**;

In the third step, the column (*IX*) analysis results in **Table 5** show that the regression coefficient of irrelevant technological diversification and firm independent innovation is 0.349 and significant at a 1% level. **H6** is supported. The coefficient of R&D human resource slack on firm independent innovation at this time is 0.941 and significant at a 1% level. The above results suggest that irrelevant technological diversification may partially mediate the positive correlation between R&D human resource slack and firm independent innovation. However, the second step is not significant according to the results of the Winn test, so the mediating role of irrelevant technological diversification is further verified using the Sobel test or Bootstrap method. The results show that it failed the Sobel test with 10,000 Bootstrap samples, which indicates that the mediating role of irrelevant technological diversification is not valid. Thus, **H7** is **not** supported.

The reasons for this are hypothesized to be, first, the difficulty of reconfiguring R&D human resource slack in the short term due to its strong "stickiness" as it has been absorbed, and hence the difficulty of exploring unrelated technology areas. Second, diversification into unrelated technologies requires a multidisciplinary search for knowledge, which is limited by the span of R&D personnel's expertise and the fact that they can only fit into existing strategies [53], and thus difficult to cross over to unrelated technology domains. For example, a significant dilemma that firms attempting to transition into large model training technologies like ChatGPT may face is that their existing R&D staff, regardless of their functional background, lacks the skills required to meet the demands of this newest area of AI. Third, the resource rigidity associated with slack human resources [54] may also push firms to explore unrelated technology areas at their own expense, thus posing a threat to the enhancement of firm independent innovation capabilities.

**4.3.4 Examining the moderating role of TMT functional heterogeneity.** The test of the moderating role of the TMT functional heterogeneity (*FH*) in the relationship between the slack of research and development human resources (RHRS) and the firm's innovation (Inno) is shown in **Table 6**. In particular, *column (X)* is a regression with the inclusion of the independent variable R&D human resources slack and the control variables on the dependent variable firm independent innovation. *Column (XI)* (i.e., Model 6) is the result after including the moderating variable TMT functional heterogeneity and its interaction term with R&D human resource slack (FH × RHRS) into the model (1) to verify the moderating effect of TMT functional heterogeneity (FH) in the relationship between R&D human resource slack (RHRS) and firm independent innovation (Inno).

The results show that the regression coefficient of the interaction term between TMT functional heterogeneity and R&D human resource slack (Difjob × HR) and firms' independent innovation (Inno) in *column (XI)* is 6.587 and significant at a 1% level. This indicates that the TMT functional heterogeneity enhances the positive relationship between R&D human resource slack and firm independent innovation, and thus, **H8** is supported.

The moderating effect of TMT functional heterogeneity is shown in **Fig 2**.

As can be seen from **Fig 2**, the positive relationship between R&D human resource slack and firm independent innovation becomes steeper under the moderating effect of the TMT functional heterogeneity, which exacerbates the positive relationship of R&D human resource slack on firm independent innovation.

## 4.4 Robustness test

**4.4.1 Robustness tests for replacement variables.** In this section, we use the moving average of the last three years (mvInno) to replace the value of the firms' independent innovation capability for the robustness test, and the regression results are shown in **Table 7**.

*Columns (XII)*, *(XIII)*, and *(XIV)* are robustness tests for the role of related technological diversification in mediating the relationship between R&D human resource slack and firm independent innovation using a three-step approach, respectively.

*Column (XV)* is a robustness test of the moderating role of TMT functional heterogeneity in the relationship between R&D human resource slack and firm independent innovation.

The results below were generally consistent with previous analyses.

**4.4.2 Robustness tests for annual industry cross-multiplier double fixed effects.** To avoid macro-environmental effects on the whole industry, we simultaneously control for both industry and vintage effects [55]. The results of the double-fixed effects are displayed in **Table 8**.

**Table 6. Moderating effect test results of TMT functional heterogeneity.**

| Model variable name | (X)<br>Inno | (XI)<br>M6 Inno |
|---|---|---|
| Size | 0.446*** | 0.445*** |
| | (0.018) | (0.018) |
| FirmAge | -0.025 | -0.023 |
| | (0.059) | (0.059) |
| Lev | -0.105 | -0.099 |
| | (0.102) | (0.102) |
| Growth | -0.016 | -0.017 |
| | (0.049) | (0.049) |
| SOE | 0.120*** | 0.117*** |
| | (0.044) | (0.044) |
| TobinQ | 0.076*** | 0.073*** |
| | (0.014) | (0.014) |
| Board | 0.483*** | 0.490*** |
| | (0.101) | (0.100) |
| Top10 | 0.568*** | 0.564*** |
| | (0.118) | (0.118) |
| Indep | 1.172*** | 1.162*** |
| | (0.336) | (0.335) |
| Balance | 0.034 | 0.034 |
| | (0.024) | (0.024) |
| Dual | 0.038 | 0.040 |
| | (0.032) | (0.032) |
| Mhold | 0.256*** | 0.263*** |
| | (0.082) | (0.082) |
| RHRS | 0.970*** | 0.970*** |
| | (0.159) | (0.158) |
| FH×RHRS | | 6.587*** |
| | | (1.836) |
| FH | | -0.073 |
| | | (0.183) |
| _cons | -8.851*** | -8.830*** |
| | (0.454) | (0.453) |
| adj. $R^2$ | 0.303 | 0.305 |
| YEAR FE | Yes | Yes |
| IND FE | Yes | Yes |
| F | 81.465 | 72.678 |
| N | 4672 | 4672 |

Note:

* indicates p < 0.100

** indicates p < 0.050

*** indicates p < 0.010.

*Columns (XVI), (XVII),* and *(XVIII)* are robust tests of the mediating role of related technological diversification in the relationship between R&D human resource slack and firm independent innovation using a three-step approach, respectively; *column (XIX)* is a robust test of

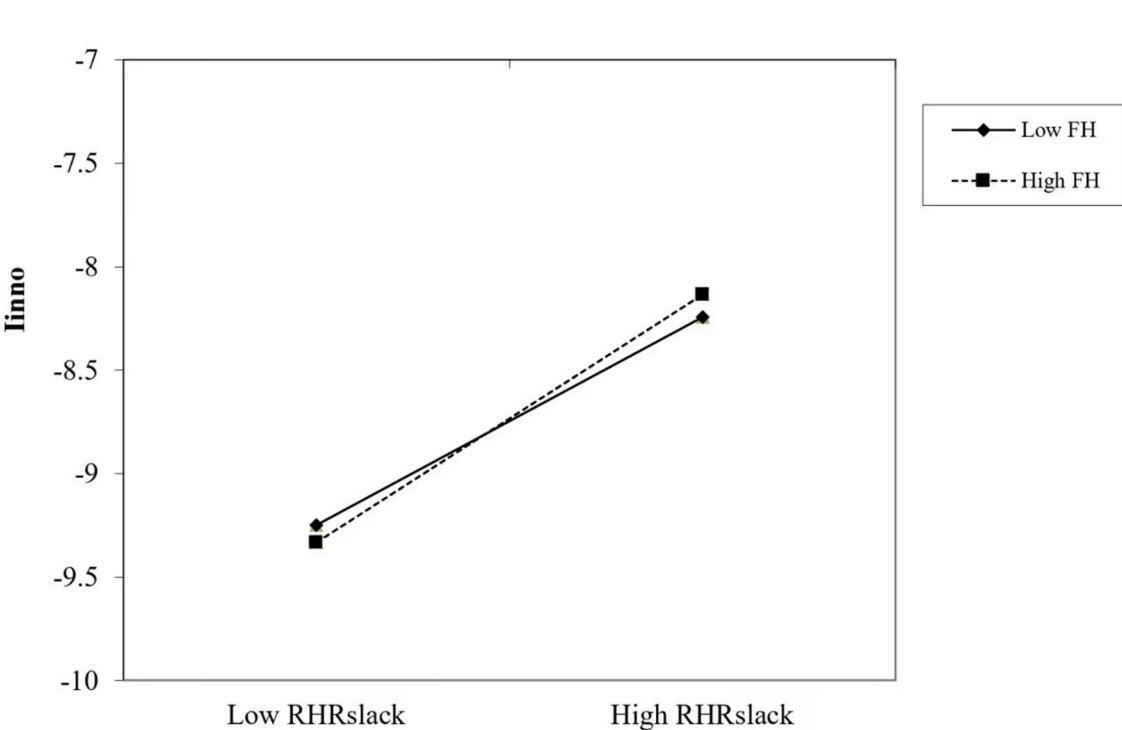

**Fig 2. The moderating effect of TMT functional heterogeneity.**

the moderating role of TMT functional heterogeneity in the relationship between R&D human resource slack and firm independent innovation. The results of the study are consistent with the previous analysis.

## 5. Conclusions and discussion

### 5.1 Conclusions

Based on theoretical and empirical analysis, we find a significant positive correlation between R&D human resource slack and firms' independent innovation. Related technological diversification plays a mediating role between them, while unrelated technological diversification plays in a non-significant way. Top management team functional background heterogeneity enhances the promotional effect of R&D human resource slack on the firms' independent innovation.

### 5.2 Discussion

To summarize, research on the impact of slack resources on firms' independent innovation is still limited. The results of this paper provide empirical evidence of the impact of R&D human resource slack on firm independent innovation. Firms should consider not only the quantity of slack resource allocation but also the direction and structure of slack resource allocation. These require firms to explore the value of slack resources for firm innovation from a deeper level.

The study of the impact of slack resources on firms' independent innovation should be anchored in the unique characteristics of independent innovation. On this basis, this study has

**Table 7. Robustness tests of the mediating role of technological diversification I.**

| Variable name | (XII) | (XIII) | (XIV) | (XV) |
|---|---|---|---|---|
| | mvInno | RTD | mvInno | mvInno |
| Size | 0.470$^{***}$ | 0.200$^{***}$ | 0.332$^{***}$ | 0.467$^{***}$ |
| | (0.024) | (0.006) | (0.024) | (0.023) |
| FirmAge | 0.101 | -0.019 | 0.107 | 0.108 |
| | (0.088) | (0.022) | (0.082) | (0.087) |
| Lev | -0.076 | 0.095$^{**}$ | -0.165 | -0.068 |
| | (0.140) | (0.039) | (0.131) | (0.140) |
| Growth | -0.163$^{**}$ | -0.012 | -0.190$^{***}$ | -0.163$^{**}$ |
| | (0.077) | (0.019) | (0.072) | (0.077) |
| SOE | 0.111$^{**}$ | 0.076$^{***}$ | 0.069 | 0.111$^{**}$ |
| | (0.055) | (0.016) | (0.053) | (0.055) |
| TobinQ | 0.072$^{***}$ | -0.008$^{*}$ | 0.083$^{***}$ | 0.069$^{***}$ |
| | (0.018) | (0.005) | (0.018) | (0.018) |
| Board | 0.405$^{***}$ | 0.135$^{***}$ | 0.244$^{**}$ | 0.425$^{***}$ |
| | (0.127) | (0.035) | (0.122) | (0.128) |
| Top10 | 0.685$^{***}$ | -0.174$^{***}$ | 0.781$^{***}$ | 0.659$^{***}$ |
| | (0.164) | (0.042) | (0.159) | (0.164) |
| Indep | 0.675 | 0.098 | 0.664 | 0.654 |
| | (0.437) | (0.117) | (0.418) | (0.433) |
| Balance | 0.009 | -0.026$^{***}$ | 0.033 | 0.009 |
| | (0.032) | (0.008) | (0.032) | (0.032) |
| Dual | 0.073 | 0.007 | 0.072$^{*}$ | 0.069 |
| | (0.044) | (0.012) | (0.042) | (0.045) |
| Mhold | 0.284$^{**}$ | 0.084$^{***}$ | 0.212$^{*}$ | 0.298$^{**}$ |
| | (0.118) | (0.030) | (0.114) | (0.118) |
| RHRS | 1.077$^{***}$ | 1.037$^{***}$ | 0.324 | 1.024$^{***}$ |
| | (0.208) | (0.059) | (0.207) | (0.204) |
| RTD | | | 0.702$^{***}$ | |
| | | | (0.054) | |
| FH×RHRS | | | | 7.861$^{***}$ |
| | | | | (2.333) |
| FH | | | | -0.418 |
| | | | | (0.280) |
| _cons | -9.316$^{***}$ | -4.624$^{***}$ | -6.012$^{***}$ | -9.294$^{***}$ |
| | (0.575) | (0.149) | (0.612) | (0.573) |
| adj. $R^2$ | 0.358 | 0.448 | 0.413 | 0.361 |
| YEAR FE | Yes | Yes | Yes | Yes |
| IND FE | Yes | Yes | Yes | Yes |
| F | 53.038 | 192.814 | 71.426 | 48.374 |
| N | 1987 | 4672 | 1987 | 1987 |

Note:

* indicates p < 0.100

** indicates p < 0.050

*** indicates p < 0.010.

distilled them into at least three characteristics: subjectivity, 'stress-resistant', and a large number of breakthroughs in basic scientific research. These are reflected in the control of all key

**Table 8. Robustness test of the mediating role of technological diversification II.**

| Variable name | (XVI) Inno | (XVII) RTD | (XVIII) Inno | (XIX) Inno |
|---|---|---|---|---|
| Size | 0.446*** | 0.199*** | 0.307*** | 0.445*** |
| | (0.018) | (0.006) | (0.019) | (0.018) |
| FirmAge | -0.024 | -0.018 | -0.012 | -0.022 |
| | (0.060) | (0.022) | (0.058) | (0.059) |
| Lev | -0.102 | 0.100** | -0.172* | -0.096 |
| | (0.104) | (0.040) | (0.099) | (0.104) |
| Growth | -0.016 | -0.006 | -0.012 | -0.018 |
| | (0.050) | (0.019) | (0.048) | (0.050) |
| SOE | 0.122*** | 0.076*** | 0.069 | 0.119*** |
| | (0.045) | (0.016) | (0.043) | (0.045) |
| TobinQ | 0.078*** | -0.011** | 0.086*** | 0.076*** |
| | (0.015) | (0.005) | (0.014) | (0.015) |
| Board | 0.486*** | 0.131*** | 0.395*** | 0.493*** |
| | (0.101) | (0.036) | (0.099) | (0.101) |
| Top10 | 0.564*** | -0.175*** | 0.686*** | 0.559*** |
| | (0.119) | (0.042) | (0.114) | (0.119) |
| Indep | 1.197*** | 0.071 | 1.148*** | 1.188*** |
| | (0.338) | (0.118) | (0.329) | (0.337) |
| Balance | 0.031 | -0.026*** | 0.049** | 0.030 |
| | (0.024) | (0.008) | (0.024) | (0.024) |
| Dual | 0.039 | 0.006 | 0.035 | 0.041 |
| | (0.032) | (0.012) | (0.031) | (0.032) |
| Mhold | 0.253*** | 0.080*** | 0.197** | 0.260*** |
| | (0.083) | (0.030) | (0.080) | (0.083) |
| RHRS | 0.976*** | 1.052*** | 0.243 | 0.976*** |
| | (0.160) | (0.059) | (0.160) | (0.159) |
| RTD | | | 0.697*** | |
| | | | (0.041) | |
| FH×RHRS | | | | 6.584*** |
| | | | | (1.861) |
| FH | | | | -0.059 |
| | | | | (0.187) |
| _cons | -8.864*** | -4.579*** | -5.671*** | -8.844*** |
| | (0.459) | (0.151) | (0.487) | (0.457) |
| adj. $R^2$ | 0.297 | 0.441 | 0.342 | 0.298 |
| YEAR FE | Yes | Yes | Yes | Yes |
| IND FE | Yes | Yes | Yes | Yes |
| F | 80.354 | 188.633 | 106.469 | 71.715 |
| N | 4669 | 4669 | 4669 | 4669 |

Note:

* indicates p < 0.100

** indicates p < 0.050

*** indicates p < 0.010.

links in the process of independent innovation. From the availability of innovation resources, the slack resources in the organization, especially the slack resources for innovation, are

precisely the important controllable resources for independent innovation. At this time, slack resources will play an important role in the survival of basic scientific research. Therefore, it is of prominent theoretical significance for the existing research to extend to a deeper level the two fields of slack resources and independent innovation, and to explore the correlation between the two on this basis.

## 5.3 Theoretical implications

Firstly, In the related research in the field of independent innovation, the antecedent influencing factors are mainly carried out from the perspective of the external environment and external stakeholders, little attention is paid to the influence of the important internal resources of the firm; and the exploration of the improvement initiatives is also mostly from the perspective of external resources. This paper responds to the resource-based theory and explores the unique impact of the slack resources of innovation function on the independent innovation of firms from the perspective of internal resources of firms, which expands the research on the pre-influencing factors of independent innovation of firms and promotes the improvement of theories in the field of independent innovation.

There is still no consensus on the impact of slack resources on firms' innovation, being positive or negative, or the U-shaped relationship. Therefore, rather than viewing slack resources as a purely positive or negative resource. By delving into more granular, function-specific slack resources and combining them with studies of specific economic environments, it is possible to untangle seemingly contradictory findings and draw stable empirical conclusions.

Secondly, given that the direction of firm independent innovation is mainly decided by TMT, and TMT also coordinates the slack resource utilization of firms, the characteristics of TMT members and their behavioral decisions often play a crucial role in the sustainable innovation development of firms. This paper combines the resource-based theory with the upper-echelon theory to incorporate the TMT functional heterogeneity into the model. Starting from the top echelon theory, the TMT functional heterogeneity is incorporated into the original research model, and the moderating role of the TMT functional heterogeneity in the relationship between the slack of R&D human resources and the firms' independent innovation is examined.

## 5.4 Practical implications

First, with the strategic vision of the firm's independent innovation, managers should have both present and long-term orientations and balance interests between them through the use of R & D human resource slack to carry out more innovative attempts. This will help to find an independent innovation mode that suits the firms' actual situation. For example, to substitute 'big and comprehensive' projects with 'specialized and refined' independent innovation mode.

Next, it should be recognized that a firm's innovation activities do not start from scratch but are premised on its ability to identify crucial internal resource endowments and utilize them effectively. From the perspective of the innovation function of slack resources, innovative slack resources are the vital endowment for technological R&D breakthroughs of firms and can be dug out at a more granular level to realize the unique value of R&D human resource slack in terms of knowledge sharing and new knowledge creation, technological research and scientific and technological rent generation. Therefore, on the one hand, managers should overcome the overconfidence and complacency that may be triggered by excessively slack resources, which may lead to inefficiency and waste of resources; on the other hand, they should create a friendly incentive atmosphere for R&D personnel through innovative management methods, dispel their concerns before investing in R&D, and motivate their creativity

and enthusiasm, to make them invest in R&D activities, thus to enhance the independent innovation ability of firms, instead of simply and roughly laying off human resources in the face of crisis.

In addition, managers should pay attention to the knowledge base structure of R&D human resource slack and its limitations to optimize it for independent innovation. Slack R&D human resources will bring the related knowledge and capabilities needed for technological diversification to the firm's independent innovation, but it should be noted that, on the other hand, slack R&D human resource has a knowledge span, thus adding difficulties to independent innovation through unrelated technological diversification. Therefore, oriented toward firm independent innovation, cross-technology competence should be trained and assessed as an essential indicator of R&D human resources slack. Moreover, since independent innovation is a process of continuous combination between knowledge elements, it is necessary to form a different knowledge base structure, which can effectively promote organizational learning and innovation.

Lastly, firms should stress corporate governance at the top level through the construction and management of the top management team, making strategic decisions to support the long-term development of firms based on the clarity of their respective powers and responsibilities. As a result, the individual expertise of the top management team members may realize the firms' mode of independent innovation.

## 5.5 Limitations and future research

Firstly, this study constructs the impact of R&D human resource slack on firm independent innovation and may ignore the value of other types of slack resources. Therefore, subsequent research may explore more slack resource variables related to firm independent innovation to further enrich the findings of this study from both theoretical and empirical aspects.

Secondly, the empirical research sample of this study is limited to manufacturing firms, and future research can further expand and refine the scope of sample selection in other industries or combine several vital industries to verify and develop the theoretical model of this study.

Thirdly, although this paper introduces the moderating variable of TMT functional heterogeneity, there are many other important boundary conditions. For example, future studies could emphasize the important role of digital technology [56].

## Supporting information

**S1 Data.**
(XLSX)

## Author Contributions

**Conceptualization:** Huijuan Li.

**Data curation:** Huijuan Li.

**Formal analysis:** Huijuan Li.

**Investigation:** Huijuan Li.

**Methodology:** Huijuan Li.

**Project administration:** Huijuan Li, Yong Wang.

**Resources:** Huijuan Li.

**Software:** Huijuan Li.

**Supervision:** Yang Li, Yong Wang.

**Writing – original draft:** Huijuan Li.

**Writing – review & editing:** Huijuan Li, Yinfei Zhao, Yong Wang.

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
