## [Decision Letter · Decision Letter 0]

7 Nov 2023

PONE-D-23-34698Where There Are Intellectuals, There Is Hope!

R&D Human Resource Slack, Technological Diversification and Independent InnovationPLOS ONE

Dear Dr. Wang,

Thank you for submitting your manuscript to PLOS ONE. After careful consideration, we feel that it has merit but does not fully meet PLOS ONE’s publication criteria as it currently stands. Therefore, we invite you to submit a revised version of the manuscript that addresses the points raised during the review process.

We look forward to receiving your revised manuscript.

Kind regards,

Jitendra Yadav, Ph.D.

Academic Editor

PLOS ONE

Journal Requirements:

3. Please amend your authorship list in your manuscript file to include authors Yong Wang, Huijuan Li, Yinfei Zhao, and Hua Huang.

Reviewers' comments:

Reviewer's Responses to Questions

**Comments to the Author**

1. Is the manuscript technically sound, and do the data support the conclusions?

Reviewer #1: Partly

Reviewer #2: Yes

2. Has the statistical analysis been performed appropriately and rigorously? 

Reviewer #1: Yes

Reviewer #2: No

3. Have the authors made all data underlying the findings in their manuscript fully available?

Reviewer #1: No

Reviewer #2: No

4. Is the manuscript presented in an intelligible fashion and written in standard English?

Reviewer #1: Yes

Reviewer #2: Yes

5. Review Comments to the Author

Reviewer #1: The topic “Where There Are Intellectuals, There Is Hope! R&D Human Resource Slack, Technological Diversification and Independent Innovation” is worthy of investigation. However, the following needs to be addressed. There are minor and major issues that should be corrected. I believe the paper could be further strengthened by added information about.

What is the purpose of the paper? What are the research implications?

Please reorganize the manuscript at the journal request. Please change the reference format.

The language of this manuscript is very bad and needs help from native speakers.

The title of the manuscript should fully demonstrate the content of this study and the relevant subjects.

Abstracts should include the purpose and findings of the study. Please simplify the presentation of the study process.

Introduction. This section should explain the study's context and research objective. Furthermore, the research gap needs to be narrowed after analyzing the previous studies. The research method is not adequately explained in the first section.

Here author must build research gap following the previous studies.-The manuscript does not answer the following concerns: Why is it timeliness to explore such a study? What makes this study different from the previously published studies? Are there any similarly findings in line with the previously published studies? Are the findings different from prior academic studies that were conducted elsewhere, if any? What are the new technologies, some recent issue highlights the importance. See: An adoption-implementation framework of digital green knowledge to improve the performance of digital green innovation practices for industry 5.0. Journal of Cleaner Production, https://doi.org/10.1016/j.jclepro.2022.132608

Incentive Mechanism for the Development of Rural New Energy Industry: New Energy Enterprise–Village Collective Linkages considering the Quantum Entanglement and Benefit Relationship. International Journal of Energy Research, vol. 2023, Article ID 1675858, 19 pages

Enhancing building energy efficiency: Formation of a cooperative digital green innovation atmosphere of photovoltaic building materials based on reciprocal incentives. AIMS Energy, 11(4), 694-722.

Methodology: Model.. I suggest authors here build your main heading on Research and data methodology. Clearly explain the model building process, and what previous studies have used similar models (model testing approach).

There is no flow in the text. It partly depends on the lack of proofreading but also on the fact that many statements and claims are made without being followed up by a clear and logical discussion. It is especially problematic in the Introduction that brings up a number of findings from different areas without linking them together.

Please make sure your conclusions' section underscores the scientific value-added of your paper, and/or the applicability of your findings/results. Highlight the novelty of your study. In addition to summarizing the actions taken and results, please strengthen the explanation of their significance. It is recommended to use quantitative reasoning comparing with appropriate benchmarks, especially those stemming from previous work.

More importantly, the choice of the variables should be explained in light of the theory and the prior literature on the topic. The arguments are simply relationships and causes very close to the replication of many studies dealing with the same thing. Seen: New Energy-Driven Construction Industry: Digital Green Innovation Investment Project Selection of Photovoltaic Building Materials Enterprises Using an Integrated Fuzzy Decision Approach, https://doi.org/10.3390/systems11010011

The authors should emphasize the important role of digital technology in future research. Some recent issue highlights the importance: The Interaction Mechanism and Dynamic Evolution of Digital Green Innovation in the Integrated Green Building Supply Chain. Systems 2023, 11, 122. https://doi.org/10.3390/systems11030122.

Discussion needs to be a coherent and cohesive set of arguments that take us beyond this study in particular, and help us see the relevance of what authors have proposed. Authors should create an independent “Discussion” section. Author need to contextualize the findings in the literature, and need to be explicit about the added value of your study towards that literature. Also other studies should be cited to increase the theoretical background of each of the method used. Findings should be contextualized in the literature and should be explicit about the added value of the study towards the literature.

As any emprical study that use different approaches I would like to ask to introduce in the Conclusion section at least a paragraph containing the study limitations. I noticed some things in the paper but a synthesis of statements related to how the study is useful (or partially useful, since are required certain further analysis) and helps potential interested readers does not really exist. Maybe in addition to the last section of Conclusion it is beneficial to introduce a section called: Discussion..

Reviewer #2: Thank you for the opportunity to review your paper.

The research paper discusses how internal human resource R & D slack can be used to develop independent innovation capabilities. It also explores the role of top management team’s functional heterogeneity and the organization’s technological diversification. The study is relevant to manufacturing industry facing an uncertain environment. The study is interesting. However, some areas of improvement are given below:

Introduction

1. It is unclear whether the slack literature refers to human resource slack. If it is not, past literature on human resource slack needs to be included, specifically in R & D area.

2. Elaboration is needed on how is independent innovation different from the regular innovation activities and how does an enterprise manage it differently

3. The research gap needs further clarification

Theoretical basis and research hypotheses

4. Independent innovation needs to be defined in contrast to standard innovation

5. Existing literature on the implication of human resource slack on innovation is missing.

6. It is not clear why human resource R & D slack will promote independent innovation specifically, and not innovation in general.

7. Relevant literature on technological diversification and TMT functional heterogeneity needs to be added. Also, the respective mediation and moderation hypotheses needs to have stronger arguments

Research design

8. The argument on choosing patents as a measure of independent innovation needs to be more convincing. Standard innovation may also use the same measure.

9. It needs to be clarified whether you are doing correlational or causal analysis. The hypotheses need to be modified accordingly

10. Please include how you have accounted for long innovation cycles and time taken for patenting. It is not necessary that the patents are obtained in the same year in which the R & D slack exists. Also, the human resources may not come with an innovation within a year. Same holds for the intervening and control variables

11. You have mentioned that there are restrictions for making the data fully available. Please provide reasons for the same.

Conclusion and Discussion

12. The theoretical implications need to indicate how the results are contributing to existing literature related to the variables and theories. Currently, it seems to be an extension of the conclusion section. The theoretical implications are not apparent.

13. The conclusion should follow the discussion as a separate section

6. PLOS authors have the option to publish the peer review history of their article (what does this mean?). If published, this will include your full peer review and any attached files.

Reviewer #1: No

Reviewer #2: No

---

## [Author Response · Author response to Decision Letter 0]

12 Jan 2024

Detailed response to specific comments of Reviewer 1

We wish to thank R1 for their thoughtful suggestions and constructive peer-review work. 

Reviewer 1, comment:

The title of the manuscript should fully demonstrate the content of this study and the relevant subjects.

Abstracts should include the purpose and findings of the study. Please simplify the presentation of the study process.

Response:

Thank R1 for underlining these deficiencies. 

Given that this study mainly focuses on the internal R&D human resource slack and its distinctive effect on independent innovation under uncertain environment, we changed the title into ‘Turning inward in difficulties: R&D human resource slack, technological diversification, and independent innovation’ to demonstrate the content of this study and the relevant subjects. 

We rewrite the abstract to make it more precise and clearer according to the comments. Specifically, it highlights the purpose, the research gap and findings. The revised abstract is as follows, and it may also be found in the Revised Manuscript with Track Changes.

Abstract: Independent innovation emphasizes the self-reliance and control of all key links. Slack resources within an organization, especially for innovation, are the critical resources that are controllable for independent innovation. However, existing research still lacks the evidence on the areas of slack innovation resources and independent innovation for deeper exploration. This research addresses this gap by providing an empirical analysis of the relationship between R&D human resource slack and firms’ independent innovation. Based on the unbalanced panel data of China’s listed manufacturing firms for eleven years, this research explores the effects of R&D human resource slack on firms’ independent innovation, the mediating mechanism of technological diversification, and the boundary effects of top management team functional heterogeneity. The results reveal that R&D human resource slack positively affects firms’ independent innovation; R&D human resource slack can promote firms’ independent innovation through related technological diversification, while the mediating effect of unrelated technological diversification is not statistically significant; the top management team functional heterogeneity strengthens the positive impact of R&D human resource slack on firm independent innovation.

Reviewer 1, comment #1:

Introduction. This section should explain the study's context and research objective. Furthermore, the research gap needs to be narrowed after analyzing the previous studies. The research method is not adequately explained in the first section.

Here author must build research gap following the previous studies.-The manuscript does not answer the following concerns: Why is it timeliness to explore such a study? What makes this study different from the previously published studies? Are there any similarly findings in line with the previously published studies? Are the findings different from prior academic studies that were conducted elsewhere, if any?

Response:

We are grateful for all R1’s comments and suggestions. We rewrite the introduction part to make it more theoretically logic. This time, as suggested, we firstly provide the policy and practical context, and then introduce the research objective more naturally. In the following paragraphs, we analyze previous literature and get research gaps. We also point out the necessities and values of this research. For the space limit, R1 may find the revised version of introduction in the resubmitted manuscript. Exact changes in the manuscript are also presented in red font. We hope it would be clearer and in accordance with the reviewer concerns. 

Reviewer 1, comment #2:

Methodology: Model.. I suggest authors here build your main heading on Research and data methodology. Clearly explain the model building process, and what previous studies have used similar models (model testing approach).

There is no flow in the text. It partly depends on the lack of proofreading but also on the fact that many statements and claims are made without being followed up by a clear and logical discussion. It is especially problematic in the Introduction that brings up a number of findings from different areas without linking them together.

Response:

First, thank R1 for your specialized comments. We agree with R1 to build our main heading on Research and data methodology. In the part of Research models, we explain the model building process by adding ‘This study adopts quantitative empirical analysis of relevant data of listed firms in the manufacturing industry, mainly through correlation analysis, regression analysis and panel multiple regression analysis to test the relationship between R&D human resource slack and firm independent innovation. Referring to Zhou et al. (2021), we construct the following high-dimensional fixed effect models to test our research hypotheses’. As for the flow in the text, especially in the Introduction, we delete unnecessary literature and add related literature to make the discussion clearer and more logical. For more details, R1 may find them in the revised manuscript. 

Lastly, we really expect our revisions would be clearer and in accordance with the reviewer concerns.

Reviewer 1, comment #3:

Please make sure your conclusions' section underscores the scientific value-added of your paper, and/or the applicability of your findings/results. Highlight the novelty of your study. In addition to summarizing the actions taken and results, please strengthen the explanation of their significance. It is recommended to use quantitative reasoning comparing with appropriate benchmarks, especially those stemming from previous work. 

Discussion needs to be a coherent and cohesive set of arguments that take us beyond this study in particular, and help us see the relevance of what authors have proposed. Authors should create an independent “Discussion” section. Author need to contextualize the findings in the literature, and need to be explicit about the added value of your study towards that literature. Also other studies should be cited to increase the theoretical background of each of the method used. Findings should be contextualized in the literature and should be explicit about the added value of the study towards the literature.

As any emprical study that use different approaches I would like to ask to introduce in the Conclusion section at least a paragraph containing the study limitations. I noticed some things in the paper but a synthesis of statements related to how the study is useful (or partially useful, since are required certain further analysis) and helps potential interested readers does not really exist. Maybe in addition to the last section of Conclusion it is beneficial to introduce a section called: Discussion..

Response:

First, we wish to thank R1 for the valuable advice. According to the R1’s suggestion, we supplemented information such as scientific value-added, applicability of our findings. More importantly, we set Discussion as a separate section, where the complemented details are ‘Discussion: To summarize, research on the impact of slack resources on firms’ independent innovation is still limited. The results of this paper provide empirical evidence of the impact of R&D human resource slack on firm independent innovation. Firms should consider not only the quantity of slack resource allocation, but also the direction and structure of slack resource allocation. These require firms to explore the value of slack resources for firm innovation from a deeper level. 

The study of the impact of slack resources on firms’ independent innovation should be anchored in the unique characteristics of independent innovation. On this basis, this study has distilled them into at least three characteristics: subjectivity, ‘stress-resistant’, and a large number of breakthroughs in basic scientific research. These are reflected in the control of all key links in the process of independent innovation. From the availability of innovation resources, the slack resources in the organization, especially the slack resources for innovation, are precisely the important controllable resources for independent innovation. At this time, slack resources will play an important role in the survival of basic scientific research. Therefore, it is of prominent theoretical significance for the existing research to extend to a deeper level the two fields of slack resources and independent innovation, and to explore the correlation between the two on this basis.’

We wish that our response and the revised manuscript have successfully addressed the concerns R1 raised. 

Reviewer 1, comment #4:

More importantly, the choice of the variables should be explained in light of the theory and the prior literature on the topic. The arguments are simply relationships and causes very close to the replication of many studies dealing with the same thing. Seen: New Energy-Driven Construction Industry: Digital Green Innovation Investment Project Selection of Photovoltaic Building Materials Enterprises Using an Integrated Fuzzy Decision Approach, https://doi.org/10.3390/systems11010011

Response:

First, we are very grateful to the reviewer 1 for pointing out this problem. In response to R1’s comments, we supplemented theory and literature to serve as supports of arguments and variables choice. R1 may find changes in the modified version.

Reviewer 1, comment #5:

The authors should emphasize the important role of digital technology in future research. Some recent issue highlights the importance: The Interaction Mechanism and Dynamic Evolution of Digital Green Innovation in the Integrated Green Building Supply Chain. Systems 2023, 11, 122. https://doi.org/10.3390/systems11030122.implications and practical implications are not clear.

Response:

Thank R1 very much for the useful comments. We emphasize the important possible affecting factors of digital technology by adding ‘For example, future studies could emphasize the important role of digital technology (Dong et al., 2023).’ We hope that our reply and revised draft can solve the problems R1 raised. 

Response to Reviewer 2

We wish to thank R2 for going beyond the call of duty and being very meticulous in the efforts to support us in perfecting the manuscript.

Reviewer 2, comment:

The research paper discusses how internal human resource R & D slack can be used to develop independent innovation capabilities. It also explores the role of top management team’s functional heterogeneity and the organization’s technological diversification. The study is relevant to manufacturing industry facing an uncertain environment. The study is interesting. However, some areas of improvement are given below:

Response:

Thank R2 very much for taking your time to review our manuscript, and we really appreciate R2’s comments and suggestions. Under R2’s instructive guidance, we polished the manuscript including Introduction, Hypothesis development, Methodology, Conclusion and Discussion.

Reviewer 2, comment #1:

Introduction

1. It is unclear whether the slack literature refers to human resource slack. If it is not, past literature on human resource slack needs to be included, specifically in R & D area.

2. Elaboration is needed on how is independent innovation different from the regular innovation activities and how does an enterprise manage it differently

3. The research gap needs further clarification

Response:

We wish to thank R2 for going beyond the call of duty and being very meticulous in the efforts to support us in perfecting the manuscript. We revised the manuscript accordingly. For example, 

1. In the Introduction, we try to clarify and include R&D area literature specifically: ‘Innovation slack resources can be seen as a refinement of existing theories of organizational slack (Mousa & Reed, 2013). From the perspective of the innovation function of slack resources, it can be uncovered that slack resources will promote the focus of the firm on the core problem (Shahzad et al., 2016). R&D human resource slack is a subcategory of slack resources (Wang et al., 2016). Given that the R&D manpower and material inputs involved in independent innovation mostly come from within the firm, while the risks and benefits of innovation are all borne by the firm (Yu et al., 2019). It is of great significance to explore and utilize internal R&D human resource slack to promote the independent innovation capability under the constraint of external resources.’

2. We also emphasize the differences between independent innovation and common innovation: ‘As it is seen, independent innovation is proposed in order not to be constrained by others in the field of key core technologies, to get rid of excessive dependence on external resources, as well as to cope with the challenges of crisis.’ ‘Firstly, this study refines the “subjectivity” and “resistance” characteristics of independent innovation…’

3. We narrow our research gaps based on context and prior literature: ‘Therefore, on the one hand, the reality is that firms have large amounts of slack resources, while on the other hand, they face external resource constraints and high risks when innovating on their own. However, research on the impact of slack resources on firms’ independent innovation is still limited in at least two aspects. First, there is a lack of research on the unique characteristics of independent innovation and the internal resource-driven perspective. Second, existing studies have not delved into slack resources at a more granular level, specific to a particular function, and in conjunction with specific economic environments. These gaps shows that research should extend into the two areas of specific slack resources and independent innovation, and then explore them at a deeper level.’

We hope our careful revision work has improved the overall quality of our research and solved the problems you raised.

Reviewer 2, comment #2:

Theoretical basis and research hypotheses

4. Independent innovation needs to be defined in contrast to standard innovation

5. Existing literature on the implication of human resource slack on innovation is missing.

6. It is not clear why human resource R & D slack will promote independent innovation specifically, and not innovation in general.

7. Relevant literature on technological diversification and TMT functional heterogeneity needs to be added. Also, the respective mediation and moderation hypotheses needs to have stronger arguments

Response:

We wish to thank R2 for the very valuable suggestions. We modify the manuscript accordingly: 

4. R2 may find the amendments in this section: ‘Independent innovation is one of critical innovation strategies, but it has distinctive characteristics. It is informed that independent innovation is innovation activity that firms adopt to overcome technology difficulties, attain valuable R&D achievements, and gain business profits of the technology achievements by themselves (Yu et al., 2019). Puranam & Srikanth (2007) also proposed a similar idea to independent innovation. They show that experienced acquirers are better able to mitigate the damaging consequences of the loss of autonomy associated with mergers and integration. Although the authors do not explicitly refer to the concept of ‘independent innovation’, they do point to the content of the ‘subjectivity’ that independent innovation emphasizes. It was not until 2009 that the concept ‘Independent Innovation’ was explicitly proposed by scholars (Shan & Zhang, 2009). This study was published in the year following the 2008 financial crisis in the United States, reflecting the ‘resistance’ characteristic of independent innovation, which is of particular significance and value in the face of a stressful economic environment.’

5. Altaf & Shah (2017) and Mousa & Reed (2013) argue that innovation slack resources can be viewed as the stock of resources within an firm that are used for innovation functions, such as underutilized personnel dedicated to development (e.g., R&D talent pool). Innovation slack resources can be utilized and allocated so that they can be used for firm innovation activities such as developing new

---

## [Decision Letter · Decision Letter 1]

25 Jan 2024

Turning inward in difficulties: R&D human resource slack, technological diversification, and independent innovation

PONE-D-23-34698R1

Dear Dr. Wang,

We’re pleased to inform you that your manuscript has been judged scientifically suitable for publication and will be formally accepted for publication once it meets all outstanding technical requirements.

Kind regards,

Jitendra Yadav, Ph.D.

Academic Editor

PLOS ONE

Reviewers' comments:

Reviewer's Responses to Questions

**Comments to the Author**

1. If the authors have adequately addressed your comments raised in a previous round of review and you feel that this manuscript is now acceptable for publication, you may indicate that here to bypass the “Comments to the Author” section, enter your conflict of interest statement in the “Confidential to Editor” section, and submit your "Accept" recommendation.

Reviewer #1: (No Response)

Reviewer #2: All comments have been addressed

2. Is the manuscript technically sound, and do the data support the conclusions?

Reviewer #1: (No Response)

Reviewer #2: Yes

3. Has the statistical analysis been performed appropriately and rigorously? 

Reviewer #1: (No Response)

Reviewer #2: Yes

4. Have the authors made all data underlying the findings in their manuscript fully available?

Reviewer #1: (No Response)

Reviewer #2: Yes

5. Is the manuscript presented in an intelligible fashion and written in standard English?

Reviewer #1: (No Response)

Reviewer #2: Yes

6. Review Comments to the Author

Reviewer #1: The manuscript has significantly improved as compared to the previous version. Indeed, the authors tried to improve it, and the main weaknesses are solved.

Thus, in my opinion, the manuscript is recommendable for publication..

Reviewer #2: The authors have done a thorough job of incorporating all comments and have explained them satisfactorily. I congratulate them on enhancing the quality of the paper considerably based on the feedback. Wishing them good luck !

7. PLOS authors have the option to publish the peer review history of their article (what does this mean?). If published, this will include your full peer review and any attached files.

Reviewer #1: No

Reviewer #2: **Yes: **Smita Chaudhry

---

## [Editor Report · Acceptance letter]

5 Jun 2024

PONE-D-23-34698R1 

PLOS ONE

Dear Dr. Wang, 

I'm pleased to inform you that your manuscript has been deemed suitable for publication in PLOS ONE. Congratulations! Your manuscript is now being handed over to our production team.

Kind regards, 

on behalf of

Dr. Jitendra Yadav 

Academic Editor

PLOS ONE